# Sliced Gromov-Wasserstein

**Titouan Vayer**
Univ. Bretagne-Sud, CNRS, IRISA
F-56000 Vannes
`titouan.vayer@irisa.fr`

**Rémi Flamary**
Univ. Côte d'Azur, OCA, Lagrange
F-06000 Nice
`remi.flamary@unice.fr`

**Romain Tavenard**
Univ. Rennes, CNRS, LETG
F-35000 Rennes
`romain.tavenard@univ-rennes2.fr`

**Laetitia Chapel**
Univ. Bretagne-Sud, CNRS, IRISA
F-56000 Vannes
`laetitia.chapel@irisa.fr`

**Nicolas Courty**
Univ. Bretagne-Sud, CNRS, IRISA
F-56000 Vannes
`nicolas.courty@irisa.fr`

## Abstract

Recently used in various machine learning contexts, the Gromov-Wasserstein distance ($GW$) allows for comparing distributions whose supports do not necessarily lie in the same metric space. However, this Optimal Transport (OT) distance requires solving a complex non convex quadratic program which is most of the time very costly both in time and memory. Contrary to $GW$, the Wasserstein distance ($W$) enjoys several properties (*e.g.* duality) that permit large scale optimization. Among those, the solution of $W$ on the real line, that only requires sorting discrete samples in 1D, allows defining the Sliced Wasserstein ($SW$) distance. This paper proposes a new divergence based on $GW$ akin to $SW$. We first derive a closed form for $GW$ when dealing with 1D distributions, based on a new result for the related quadratic assignment problem. We then define a novel OT discrepancy that can deal with large scale distributions via a slicing approach and we show how it relates to the $GW$ distance while being $O(n \log(n))$ to compute. We illustrate the behavior of this so called Sliced Gromov-Wasserstein ($SGW$) discrepancy in experiments where we demonstrate its ability to tackle similar problems as $GW$ while being several order of magnitudes faster to compute.

## 1 Introduction

Optimal Transport (OT) aims at defining ways to compare probability distributions. One typical example is the Wasserstein distance ($W$) that has been used for varied tasks ranging from computer graphics [1] to signal processing [2]. It has proved to be very useful for a wide range of machine learning tasks including generative modelling (Wasserstein GANs [3]), domain adaptation [4] or supervised embeddings for classification purposes [5]. However one limitation of this approach is that it implicitly assumes *aligned* distributions, *i.e.* that lie in the same metric space or at least between spaces where a meaningful distance *across* domains can be computed. From another perspective, the Gromov-Wasserstein ($GW$) distance benefits from more flexibility when it comes to the more challenging scenario where heterogeneous distributions are involved, *i.e.* distributions whose supports do not necessarily lie on the same metric space. It only requires modelling the topological or relational aspects of the distributions *within* each domain in order to compare them. As such, it has

recently received a high interest in the machine learning community, solving learning tasks such as heterogenous domain adaptation [6], deep metric alignment [7], graph classification [8] or generative modelling [9].

OT is known to be a computationally difficult problem: the Wasserstein distance involves a linear program that most of the time prevents its use to settings with more than a few tens of thousands of points. For medium to large scale problems, some methods relying *e.g.* on entropic regularization [10] or dual formulation [11] have been investigated in the past years. Among them, one builds upon the mono-dimensional case where computing the Wasserstein distance can be trivially solved in $O(n \log n)$ by sorting points in order and pairing them from left to right. While this 1D case has a limited interest *per se*, it is one of the main ingredients of the *sliced* Wasserstein distance ($SW$) [12]: high-dimensional data are linearly projected into sets of mono-dimensional distributions, the sliced Wasserstein distance being the average of the Wasserstein distances between all projected measures. This framework provides an efficient algorithm that can handle millions of points and has similar properties to the Wasserstein distance [13]. As such, it has attracted attention and has been successfully used in various tasks such as barycenter computation [14], classification [15] or generative modeling [16–19].

Regarding $GW$, the optimization problem is a non-convex quadratic program, with a prohibitive computational cost for problems with more than a few thousands of points: the number of terms grows quadratically with the number of samples and one cannot rely on a dual formulation as for Wasserstein. However several approaches have been proposed to tackle its computation. Initially approximated by a linear lower bound [20], $GW$ was thereafter estimated through an entropy regularized version that can be efficiently computed by iterating Sinkhorn projections [21, 22]. More recently a conditional gradient scheme relying on linear program OT solvers was proposed in [8]. However, as discussed more in detail in Sec. 2, all these methods are still too costly for large scale scenarii.

In this paper, we propose a new formulation related to $GW$ that lowers its computational cost. To that extent, we derive a novel OT discrepancy called Sliced Gromov-Wasserstein ($SGW$). It is similar in spirit to the Sliced Wasserstein distance as it relies on the exact computation of 1D $GW$ distances of distributions projected onto random directions. We notably provide the first 1D closed form solution of the $GW$ problem by proving a new result about the Quadratic Assignment Problem (QAP) for matrices that are squared euclidean distances of real numbers. Computation of $SGW$ for discrete distributions of $n$ points is $O(L\, n \log(n))$, where $L$ is the number of sampled directions. This complexity is the same as the Sliced-Wasserstein distance and is even lower than computing the value of $GW$ which is $O(n^3)$ for a known coupling (once the optimization problem solved) in the general case [22]. Experimental validation shows that $SGW$ retains various properties of $GW$ while being much cheaper to compute, allowing its use in difficult large scale settings such as large mesh matching or generative adversarial networks.

**Notations** The simplex histogram with $n$ bins will be denoted as $\Sigma_n = \{a \in (\mathbb{R}_+)^n, \sum_i a_i = 1\}$. For two histograms $a \in \Sigma_n$ and $b \in \Sigma_m$ we note $\Pi(a, b)$ the set of all couplings of $a$ and $b$, *i.e.* the set $\Pi(a, b) = \{\pi \in \mathbb{R}_+^{n \times m} \| \sum_i \pi_{i,j} = b_j; \sum_j \pi_{i,j} = a_i\}$. $S_n$ is the set of all permutations of $\{1, ..., n\}$.

We note $\|.\|_{k,p}$ the $\ell_k$ norm on $\mathbb{R}^p$. For any norm $\|.\|$ we note $d_{\|.\|}$ the distance induced by this norm.

$\delta_x$ is the dirac measure in $x$ *s.t.* a discrete measure $\mu \in \mathcal{P}(\mathbb{R}^p)$ can be written $\mu = \sum_{i=1}^n a_i \delta_{x_i}$ with $x_i \in \mathbb{R}^p$. For a continuous map $f : \mathbb{R}^p \to \mathbb{R}^q$ we note $f\#$ its push-forward operator. $f\#$ moves the positions of all the points in the support of the measure to define a new measure $f\#\mu \in \mathcal{P}(\mathbb{R}^q)$ *s.t.* $f\#\mu \overset{\text{def.}}{=} \sum_i a_i \delta_{f(x_i)}$. We note $\mathcal{O}(p)$ the subset of $\mathbb{R}^{p \times p}$ of all orthogonal matrices. Finally $\mathbb{V}_p(\mathbb{R}^q)$ is the Stiefel manifold, *i.e.* the set of all orthonormal $p$-frames in $\mathbb{R}^q$ or equivalently $\mathbb{V}_p(\mathbb{R}^q) = \{\Delta \in \mathbb{R}^{q \times p} | \Delta^T \Delta = I_p\}$.

## 2   Gromov-Wasserstein distance

OT provides a way of inferring correspondences between two distributions by leveraging their intrinsic geometries. If one has measures $\mu$ and $\nu$ on two spaces $X$ and $Y$, OT aims at finding a correspondence (or *transport*) map $\pi \in \mathcal{P}(X \times Y)$ such that the marginals of $\pi$ are respectively $\mu$ and $\nu$. When a meaningful distance or cost $c : X \times Y \mapsto \mathbb{R}_+$ *across* the two domains can be computed, classical OT

relies on minimizing the total transportation cost between the two distributions $\int_{X \times Y} c(x, y) d\pi(x, y)$ *w.r.t.* $\pi$. The minimum total cost is often called the Wasserstein distance between $\mu$ and $\nu$ [23].

However, this approach fails when a meaningful cost *across* the distributions cannot be defined, which is the case when $\mu$ and $\nu$ live for instance in Euclidean spaces of different dimensions or more generally when $X$ and $Y$ are *unaligned*, *i.e.* when their features are not in correspondence. This is particularly the case for features learned with deep learning as they can usually be arbitrarily rotated or permuted. In this context, the $W$ distance with the naive cost $c(x, y) = \|x - y\|$ fails at capturing the similarity between the distributions. Some works address this issue by realigning spaces $X$ and $Y$ using a global transformation before using the classical $W$ distance [24]. From another perspective, the so-called $GW$ distance [25] has been investigated in the past few years and rather relies on comparing *intra*-domain distances $c_X$ and $c_Y$.

**Definition and basic properties**    Let $\mu \in \mathcal{P}(\mathbb{R}^p)$ and $\nu \in \mathcal{P}(\mathbb{R}^q)$ with $p \leq q$ be discrete measures on Euclidean spaces with $\mu = \sum_{i=1}^{n} a_i \delta_{x_i}$ and $\nu = \sum_{i=1}^{m} b_j \delta_{y_j}$ of supports $X$ and $Y$, where $a \in \Sigma_n$ and $b \in \Sigma_m$ are histograms.

Let $c_X : \mathbb{R}^p \times \mathbb{R}^p \to \mathbb{R}_+$ (*resp.* $c_Y : \mathbb{R}^q \times \mathbb{R}^q \to \mathbb{R}_+$) measures the similarity between the samples in $\mu$ (*resp.* $\nu$). The Gromov-Wasserstein ($GW$) distance is defined as:

$$GW_2^2(c_X, c_Y, \mu, \nu) = \min_{\pi \in \Pi(a,b)} J(c_X, c_Y, \pi) \tag{1}$$

where

$$J(c_X, c_Y, \pi) = \sum_{i,j,k,l} \left| c_X(x_i, x_k) - c_Y(y_j, y_l) \right|^2 \pi_{i,j} \pi_{k,l}.$$

The resulting coupling $\pi$ is a fuzzy correspondance map between the points of the distributions which tends to associate pairs of points with similar distances within each pair: the more similar $c_X(x_i, x_k)$ is to $c_Y(y_j, y_l)$, the stronger the transport coefficients $\pi_{i,j}$ and $\pi_{k,l}$ are. The $GW$ distance enjoys many desirable properties when $c_X$ and $c_Y$ are distances so that $(X, c_X, \mu)$ and $(Y, c_Y, \nu)$ are called *measurable metric spaces* (mm-spaces) [25]. In this case, $GW$ is a metric *w.r.t.* the measure preserving isometries. More precisely, it is symmetric, satisfies the triangle inequality when considering three mm-spaces, and vanishes *iff* the mm-spaces are *isomorphic*, *i.e.* when there exists a surjective function $f : X \to Y$ such that $f \# \mu = \nu$ ($f$ preserves the measures) and $\forall x, x' \in X^2, c_Y(f(x), f(x')) = c_X(x, x')$ ($f$ is an isometry). With a slight abuse of notations we will say that $\mu$ and $\nu$ are *isomorphic* when this occurs. The $GW$ distance has several interesting properties, especially in terms of invariances. It is clear from its formulation in eq. (1) that it is invariant to translations, permutations or rotations on both distributions when Euclidean distances are used. This last property allows finding correspondences between complex word embeddings between different languages [26]. Interestingly enough, when spaces have the same dimension, it has been proven that computing $GW$ is equivalent to realigning both spaces using some linear transformation and then computing the $W$ distance on the realigned measures (Lemma 4.3 in [24]).

$GW$ can also be used with other similarity functions for $c_X$ and $c_Y$ (*e.g.* kernels [22] or squared integrable functions [27]). In this work, we focus on squared euclidean distances, *i.e.* $c_X(x, x') = \|x - x'\|_{2,p}^2, c_Y(y, y') = \|y - y'\|_{2,q}^2$. This particular case is tackled by the theory of *gauged measure spaces* [20, 27] where authors generalize mm-spaces with weaker assumptions on $c_X, c_Y$ than the distance assumptions. More importantly in our context, invariants are the same as for distances since $GW$ still vanishes *iff* there exists a measure preserving isometry (*cf.* supplementary material) and the symmetry and triangle inequality are also preserved (see [20]).

**Computational aspects**    The optimization problem (1) is a non-convex Quadratic Program (QP). Those problems are notoriously hard to solve since one cannot rely on convexity and only descent methods converging to local minima are available. The problem can be tackled by solving iterative linearizations of the quadratic function with a conditional gradient as done in [8]. In this case, each iteration requires the optimization of a classical OT problem, that is $O(n^3)$. Another approach consists in solving an approximation of problem (1) by adding an entropic regularization as proposed in [22]. This leads to an efficient projected gradient algorithm where each iteration requires solving a regularized OT with the Sinkhorn algorithm that has be shown to be nearly $O(n^2)$ and implemented efficiently on GPU. Still note that even though iterations for regularized $GW$ are faster, the computation of the final cost is $O(n^3)$ [22, Remark 1].

# 3 From 1D GW to Sliced Gromov-Wasserstein

In this section, we first provide and prove a solution for an 1D Quadratic Assignement Problem (QAP) with a quasilinear time complexity of $O(n \log(n))$. This new special case of the QAP is shown to be equivalent to the *hard assignment* version of $GW$, called the Gromov-Monge ($GM$) problem, with squared Euclidean cost for distributions lying on the real line. We also show that, in this context, solving $GM$ is equivalent to solving $GW$. We derive a new discrepancy named Sliced Gromov-Wasserstein ($SGW$) that relies on these findings for efficient computation.

**Solving a Quadratic Assignement Problem in 1D**   In Koopmans-Beckmann form [28] a QAP takes as input two $n \times n$ matrices $A = (a_{ij})$, $B = (b_{ij})$. The goal is to find a permutation $\sigma \in S_n$, the set of all permutations of $\{1, \cdots, n\}$, which minimizes the objective function $\sum_{i,j=1}^{n} a_{i,j} b_{\sigma(i),\sigma(j)}$. In full generality this problem is NP-hard. However when matrices $A$ and $B$ have simple known structures, solutions can still be found (*e.g.* diagonal structure such as Toeplitz matrix or separability properties such as $a_{i,j} = \alpha_i \alpha_j$ [29–31]). We refer the reader to [32, 33] for comprehensive surveys on the QAP. The following theorem is a new result about QAP and states that it can be solved when $A$ and $B$ are squared Euclidean distance matrices of sorted real numbers:

**Theorem 3.1.** *A new special case for the Quadratic Assignment Problem*

*For real numbers $x_1 \leq ... \leq x_n$ and $y_1 \leq ... \leq y_n$,*

$$\min_{\sigma \in S_n} \sum_{i,j} -(x_i - x_j)^2 (y_{\sigma(i)} - y_{\sigma(j)})^2 \tag{2}$$

*is achieved either by the identity permutation $\sigma(i) = i$ or the anti-identity permutation $\sigma(i) = n + 1 - i$.*

To the best of our knowledge, this result is new. It states that if one wants to find the best one-to-one correspondence of real numbers such that their pairwise distances are best conserved, it suffices to sort the points and check whether the identity has a better cost than the anti-identity. Proof of this theorem can be found in the supplementary material. We postulate that this result also holds for $a_{ij} = |x_i - x_j|^k$ and $b_{ij} = -|y_i - y_j|^k$ with any $k \geq 1$ but leave this study for future works.

**Gromov-Wasserstein distance on the real line**   When $n = m$ and $a_i = b_j = \frac{1}{n}$, one can look for the *hard assignment* version of the $GW$ distance resulting in the Gromov-Monge problem [34] associated with the following $GM$ distance:

$$GM_2(c_X, c_Y, \mu, \nu) = \min_{\sigma \in S_n} \frac{1}{n^2} \sum_{i,j} \left| c_X(x_i, x_j) - c_Y(y_{\sigma(i)}, y_{\sigma(j)}) \right|^2 \tag{3}$$

where $\sigma \in S_n$ is a one-to-one mapping $\{1, \cdots, n\} \to \{1, \cdots, n\}$. Interestingly when the permutation $\sigma$ is known, the computation of the cost is $O(n^2)$ which is far better than $O(n^3)$ for the general $GW$ case. It is easy to see that this problem is equivalent to minimizing $\sum_{i,j=1}^{n} a_{i,j} b_{\sigma(i),\sigma(j)}$ with $a_{ij} = c_X(x_i, x_j)$ and $b_{ij} = -c_Y(y_{\sigma(i)}, y_{\sigma(j)})$. Thus, when a squared Euclidean cost is used for distributions lying on the real line, we exactly recover the solution of the $GM$ problem defined in eq. (2). As matter of consequence, theorem 3.1 provides an efficient way of solving the Gromov-Monge problem.

Moreover, this theorem also allows finding a closed form for the $GW$ distance. Indeed, some recent advances in graph matching state that, under some conditions on $A$ and $B$, the assignment problem is equivalent to its *soft-assignment* counterpart [35]. This way, using both Theorem 3.1 and [35], one can find a solvable case for the $GW$ distance when $p, q = 1$ as stated in the following theorem:

**Theorem 3.2.** *Closed form for GW and GM in 1D for $n = m$ and uniform weights*

*Let $\mu = \frac{1}{n} \sum_{i=1}^{n} \delta_{x_i} \in \mathcal{P}(\mathbb{R})$ and $\nu = \frac{1}{n} \sum_{i=1}^{n} \delta_{y_j} \in \mathcal{P}(\mathbb{R})$ with $\mathbb{R}$ equipped with the Euclidean distance $d(x, x') = |x - x'|$. Then $GW_2(d^2, \mu, \nu) = GM_2(d^2, \mu, \nu)$.*

*Moreover, if $x_1 \leq \cdots \leq x_n$ and $y_1 \leq \cdots \leq y_n$ this result is achieved either by the identity or the anti-identity permutation.*

*Sketch of the proof.* Since $d^2$ is conditionally negative definite of order 1 (see *e.g.* Prop 3 and 4 in [36]), one can use the theory developed in [35] to prove that the assignment problem of $GM$

is equivalent to $GW$. Note that this result is true also for $c_X(x, x') = \|x - x'\|_{2,p}^2$, $c_Y(y, y') = \|y - y'\|_{2,q}^2$ for any $p$ and $q$. Using Theorem 3.1 for the $GM$ distance concludes the proof. $\qquad\square$

A more detailed proof is provided as supplementary material. In the following, we only consider the case where $\mu$ and $\nu$ are discrete measures with the same number of atoms $n = m$, uniform weights and $p \leq q$. Note also that, while both possible solutions for problem (3) can be computed in $O(n \log(n))$, finding the best one requires the computation of the cost which seems, at first sight, to have a $O(n^2)$ complexity. However, under the hypotheses of theorem 3.2, the cost can be computed in $O(n)$. Indeed, in this case, one can develop the sum in eq (3) to compute it in $O(n)$ operations using binomial expansion (see details in the supplementary materials) so that the overall complexity of finding the best assignment and computing the cost is $O(n \log(n))$ which is the same complexity as the Sliced Wasserstein distance.

**Sliced Gromov-Wasserstein discrepancy**    Theorem 3.2 can be put in perspective with the Wasserstein distance for 1D distributions which is achieved by the identity permutation when points are sorted [37]. As explained in the introduction, this result was used to approximate the Wasserstein distance between measures of $\mathbb{R}^p$ using the so called Sliced Wasserstein (SW) distance [14]. The main idea is to project the points of the measures on lines of $\mathbb{R}^p$ where computing a Wasserstein distance is easy since it only involves a simple sort and to average these distances. It has been proven that $SW$ and $W$ are equivalent in terms of metric on compact domains [13]. In the same philosophy we build upon Theorem 3.2 to define a "sliced" version of the $GW$ distance.

Let $\mathbf{S}^{q-1} = \{\theta \in \mathbb{R}^q : \|\theta\|_{2,q} = 1\}$ be the $q$-dimensional hypersphere and $\lambda_{q-1}$ the uniform measure on $\mathbf{S}^{q-1}$. For $\theta$ we note $P_\theta$ the projection on $\theta$, *i.e.* $P_\theta(x) = \langle x, \theta \rangle$. For a linear map $\Delta \in \mathbb{R}^{q \times p}$ (identified with slight abuses of notation by its corresponding matrix), we define the Sliced Gromov-Wasserstein (SGW) as follows:

$$SGW_\Delta(\mu, \nu) = \mathop{\mathbb{E}}_{\theta \sim \lambda_{q-1}} [GW_2^2(d^2, P_\theta \# \mu_\Delta, P_\theta \# \nu)] = \fint_{\mathbf{S}^{q-1}} GW_2^2(d^2, P_\theta \# \mu_\Delta, P_\theta \# \nu) d\theta \quad (4)$$

where $\mu_\Delta = \Delta \# \mu \in \mathcal{P}(\mathbb{R}^q)$ and $\fint_{\mathbf{S}^{q-1}} = \frac{1}{\text{vol}(\mathbf{S}^{q-1})} \int_{\mathbf{S}^{q-1}}$ is the normalized integral and can be seen as the expectation for $\theta$ following a uniform distribution of support $\mathbf{S}^{q-1}$. The function $\Delta$ acts as a mapping for a point in $\mathbb{R}^p$ of the measure $\mu$ onto $\mathbb{R}^q$. When $p = q$ and when we consider $\Delta$ as the identity map we simply write $SGW(\mu, \nu)$ instead of $SGW_{I_p}(\mu, \nu)$. When $p < q$, one straightforward choice is $\Delta = \Delta_{pad}$ the "uplifting" operator which pads each point of the measure with zeros: $\Delta_{pad}(x) = (x_1, \ldots, x_p, \underbrace{0, \ldots, 0}_{q-p})$. The procedure is illustrated in Fig 1.

In general fixing $\Delta$ implies that some properties of $GW$, such as the rotational invariance, are lost. Consequently, we also propose a variant of SGW that does not depends on the choice of $\Delta$ called Rotation Invariant SGW ($RISGW$) and expressed as the following:

$$RISGW(\mu, \nu) = \min_{\Delta \in \mathbb{V}_p(\mathbb{R}^q)} SGW_\Delta(\mu, \nu). \quad (5)$$

We propose to minimize $SGW_\Delta$ with respect to $\Delta$ in the Stiefel manifold [38] which can be seen as finding an optimal projector of the measure $\mu$ [39, 40]. This formulation comes at the cost of an additional optimization step but allows recovering one key property of GW. When $p = q$ this encompasses for *e.g.* all rotations of the space, making $RISGW$ invariant by rotation (see theorem 3.3).

Interestingly enough, $SGW$ holds various properties of the $GW$ distance as summarized in the following theorem:

**Theorem 3.3.** *Properties of SGW*

- *For all $\Delta$, $SGW_\Delta$ and $RISGW$ are translation invariant. $RISGW$ is also rotational invariant when $p = q$, more precisely if $Q \in \mathcal{O}(p)$ is an orthogonal matrix, $RISGW(Q \# \mu, \nu) = RISGW(\mu, \nu)$ (same for any $Q' \in \mathcal{O}(q)$ applied on $\nu$).*

- *$SGW$ and $RISGW$ are pseudo-distances on $\mathcal{P}(\mathbb{R}^p)$, i.e. they are symmetric, satisfy the triangle inequality and $SGW(\mu, \mu) = RISGW(\mu, \mu) = 0$.*

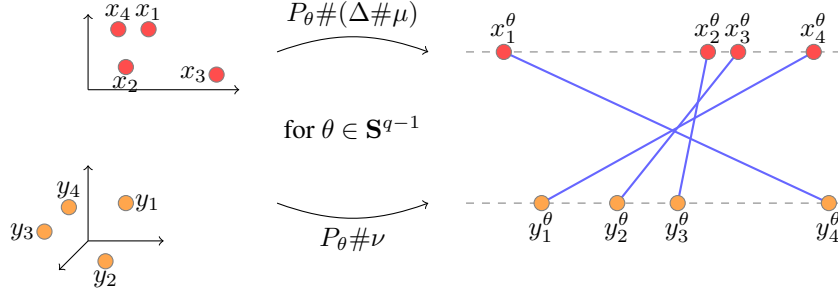

Figure 1: Example in dimension $p = 2$ and $q = 3$ (left) that are projected on the line (right). The solution for this projection is the anti-diagonal coupling.

- *For $\mu, \nu \in \mathcal{P}(\mathbb{R}^p) \times \mathcal{P}(\mathbb{R}^p)$ as defined previously, if $SGW(\mu, \nu) = 0$ then $\mu$ and $\nu$ are isomorphic for the distance induced by the $\ell_1$ norm on $\mathbb{R}^p$. In particular this implies $GW_2(d_{\|.\|_{1,p}}, \mu, \nu) = 0$.*

(with a slight abuse of notation we identify the matrix $Q$ by its linear application). A proof of this theorem can be found in the supplementary material. This theorem states that if $SGW$ vanishes then measures must be isometric, as it is the case for $GW$. It states also that $RISGW$ holds most of the properties of $GW$ in term of invariances.

**Remark**   The $\Delta$ map can also be used in the context of the Sliced Wasserstein distance so as to define $SW_\Delta(\mu, \nu)$, $RISW(\mu, \nu)$ for $\mu, \nu \in \mathcal{P}(\mathbb{R}^p) \times \mathcal{P}(\mathbb{R}^q)$ with $p \neq q$. Please note that from a purely computational point of view, complexities of these discrepancies are the same as $SGW$ and $RISGW$. Also, unlike $SGW$ and $RISGW$, these discrepancies are not translation invariant. More details are given in the supplementary material.

**Computational aspects**   Similarly to Sliced Wasserstein, $SGW$ can be approximated by replacing the integral by a finite sum over randomly drawn directions. In practice we compute $SGW$ as the average of $GW_2^2$ projected on $L$ directions $\theta$. While the sum in (4) can be implemented with libraries such as Pykeops [41], Theorem 3.2 shows that computing (4) is achieved by an $O(n \log(n))$ sorting of the projected samples and by finding the optimal permutation which is either the identity or the anti identity. Moreover computing the cost is $O(n)$ for each projection as explained previously. Thus the overall complexity of computing $SGW$ with $L$ projections is $O(Ln(p + q) + Ln \log(n) + Ln) = O(Ln(p+q+\log(n)))$ when taking into account the cost of projections. Note that these computations can be efficiently implemented in parallel on GPUs with modern toolkits such as Pytorch [42].

The complexity of solving $RISGW$ is higher but one can rely on efficient algorithms for optimizing on the Stiefel manifold [38] that have been implemented in several toolboxes [43, 44]. Note that each iteration in a manifold gradient decent requires the solution of $SGW$, that can be computed and differentiated efficiently with the frameworks described above. Moreover, the optimization over the Stiefel manifold does not depend on the number of points but only on the dimension $d$ of the problem so that overall complexity is $n_{\text{iter}}(Ln(d + \log(n)) + d^3)$, which is affordable for small $d$. In practice, we observed in the numerical experiments that RISGW converges in few iterations (the order of 10).

## 4   Experimental results

The goal of this section is to validate $SGW$ and its rotational invariant on both quantitative (execution time) and qualitative sides. All the experiments were conducted on a standard computer equipped with a NVIDIA Titan X GPU.

**SGW and RISGW on spiral dataset**   As a first example, we use the spiral dataset from sklearn toolbox and compute $GW$, $SGW$ and $RISGW$ on $n = 100$ samples with $L = 20$ sampled lines for different rotations of the target distribution. The optimization of $\Delta$ on the Stiefel manifold is performed using Pymanopt [43] with automatic differentiation with autograd [45]. Some examples of empirical distributions are available in Figure 2 (left). The mean value of $GW$, $SGW$ and $RISGW$

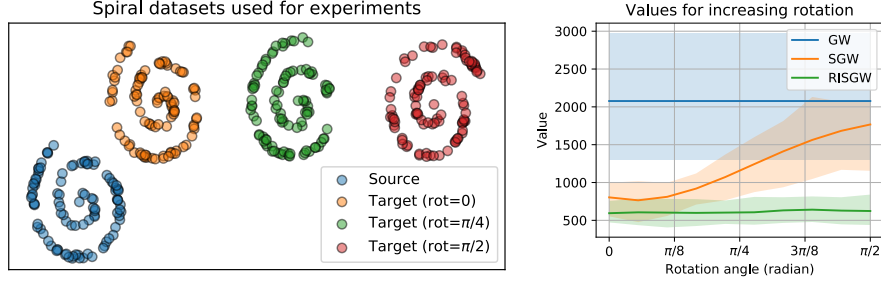

Figure 2: Illustration of $SGW$, $RISGW$ and $GW$ on spiral dataset for varying rotations on discrete 2D spiral dataset. (Left) Examples of spiral distributions for source and target with different rotations. (Right) Average value of $SGW$, $GW$ and $RISGW$ with $L = 20$ as a function of rotation angle of the target. Colored areas correspond to the 20% and 80% percentiles.

are reported on Figure 2 (right) where we can see that $RISGW$ is invariant to rotation as $GW$ whereas $SGW$ with $\Delta = I_d$ is clearly not.

**Runtimes comparison**    We perform a comparison between runtimes of $SGW$, $GW$ and its entropic counterpart [21]. We calculate these distances between two 2D random measures of $n \in \{1e2, ..., 1e6\}$ points. For $SGW$, the number of projections $L$ is taken from $\{50, 200\}$. We use the Python Optimal Transport (POT) toolbox [46] to compute $GW$ distance on CPU. For entropic-$GW$ we use the Pytorch GPU implementation from [9] that uses the log-stabilized Sinkhorn algorithm [47] with a regularization parameter $\varepsilon = 100$. For $SGW$, we implemented both a Numpy implementation and a Pytorch implementation running on GPU. Fig. 3 illustrates the results.

$SGW$ is the only method which scales *w.r.t.* the number of samples and allows computation for $n > 10^4$. While entropic-$GW$ uses GPU, it is still slow because the gradient step size in the algorithm is inversely proportional to the regularization parameter [22] which highly curtails the convergence of the method. On CPU, $SGW$ is two orders of magnitude faster than $GW$. On GPU, $SGW$ is five orders of magnitude better than $GW$ and four orders of magnitude better than entropic $GW$. Still the slope of both $GW$ implementations are surprisingly good, probably due to their maximum iteration stopping criteria. In this experiment we were able to compute $SGW$ between $10^6$ points in

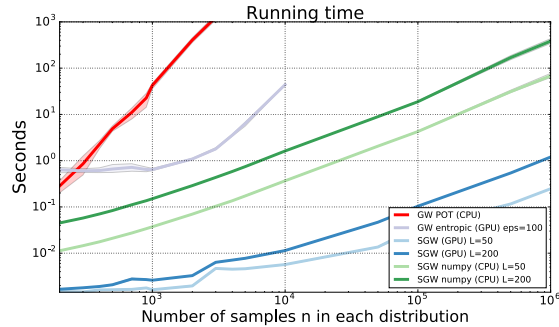

Figure 3: Runtimes comparison between $SGW$, $GW$, entropic-$GW$ between two 2D random distributions with varying number of points from 0 to $10^6$ in log-log scale. The time includes the calculation of the pair-to-pair distances.

1s. Finally note that we recover exactly a quasi-linear slope, corresponding to the $O(n \log(n))$ complexity for $SGW$.

**Meshes comparison**    In the context of computer graphics, $GW$ can be used to quantify the correspondances between two meshes. A direct interest is found in shape retrieval, search, exploration or organization of databases. In order to recover experimentally some of the desired properties of the $GW$ distance, we reproduce an experiment originally conducted in [48] and presented in [21] with the use of entropic-$GW$.

From a given time series of 45 meshes representing a galloping horse, the goal is to conduct a multi-dimensional scaling (MDS) of the pairwise distances, computed with $SGW$ between the meshes, that allows ploting each mesh as a 2D point. As one can observe in Fig. 4, the cyclical nature of this motion is recovered in this 2D plot, as already illustrated in [21] with the $GW$ distance. Each horse

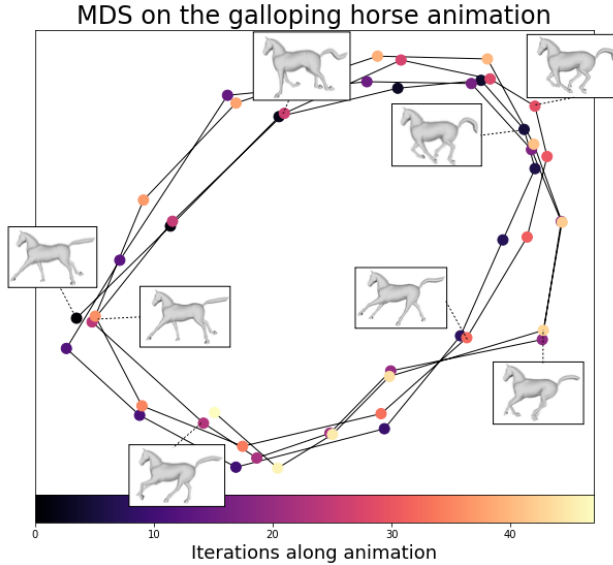

Figure 4: Each sample in this Figure corresponds to a mesh and is colored by the corresponding time iteration. One can see that the cyclical nature of the motion is recovered.

mesh is composed of approximately $9,000$ vertices. The average time for computing one distance is around 30 minutes using the POT implementation, which makes the computation of the full pairwise distance matrix impractical (as already mentioned in [21]). In contrast, our method only requires 25 minutes to compute the full distance matrix, with an average of 1.5s per mesh pair, using our CPU implementation. This clearly highlights the benefits of our method in this case.

**SGW as a generative adversarial network (GAN) loss**    In a recent paper [9], Bunne and colleagues propose a new variant of GAN between incomparable spaces, *i.e.* of different dimensions. In contrast with classical divergences such as Wasserstein, they suggest to capture the intrinsic relations between the samples of the target probability distribution by using $GW$ as a loss for learning. More formally, this translates into the following optimization problem over a desired generator $G$:

$$G^* = \arg\min GW_2^2(c_X, c_{G(Z)}, \mu, \nu_G), \tag{6}$$

where $Z$ is a random noise following a prescribed low-dimensional distribution (typically Gaussian), $G(Z)$ performs the uplifting of $Z$ in the desired dimensional space, and $c_{G(Z)}$ is the corresponding metric. $\mu$ and $\nu_G$ correspond respectively to the target and generated distributions, that we might want to align in the sense of $GW$. Following the same idea, and the fact that sliced variants of the Wasserstein distance have been successfully used in the context of GAN [17], we propose to use $SGW$ instead of $GW$ as a loss for learning $G$. As a proof of concept, we reproduce the simple toy example of [9]. Those examples consist in generating 2D or 3D distributions from target distributions either in 2D or 3D spaces (Fig. 5 and supplementary material). These distributions are formed by $3,000$ samples. We do not use their adversarial metric learning as it might confuse the objectives of this experiment and as it is not required for these low dimensional problems [9]. The generator $G$ is designed as a simple multilayer perceptron with 2 hidden layers of respectively 256 and 128 units with ReLu activation functions, and one final layer with 2 or 3 output neurons (with linear activation) as output, depending on the experiment. The Adam optimizer is used, with a learning rate of $2.10^{-4}$ and $\beta_1 = 0.5, \beta_2 = 0.99$. The convergence to a visually acceptable solution takes a few hundred epochs. Contrary to [9], we directly back-propagate through our loss, without having to explicit a coupling matrix and resorting to the envelope Theorem. Compared to [9] and the use of entropic-$GW$ , the time per epoch is more than one order of magnitude faster, as expected from previous experiment.

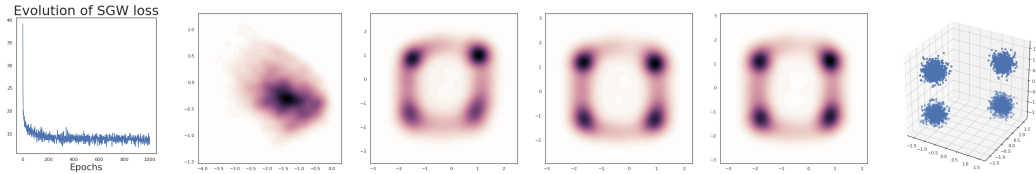

Figure 5: Using $SGW$ in a GAN loss. First image shows the loss value along epochs. The next 4 images are produced by sampling the generated distribution ($3,000$ samples, plotted as a continuous density map). Last image shows the target 3D distribution.

## 5 Discussion and conclusion

In this work, we establish a new result about Quadratic Assignment Problem when matrices are squared euclidean distances on the real line, and use it to state a closed form expression for $GW$ between monodimensional measures. Building upon this result we define a new similarity measure, called the Sliced Gromov-Wasserstein and a variant Rotation-invariant $SGW$ and prove that both conserve various properties of the $GW$ distance while being cheaper to compute and applicable in a large-scale setting. Notably $SGW$ can be computed in 1 second for distributions with 1 million samples each. This paves the way for novel promising machine learning applications of optimal transport between metric spaces.

Yet, several questions are raised in this work. Notably, our method perfectly fits the case when the two distributions are given empirically through samples embedded in an Hilbertian space, that allows for projection on the real line. This is the case in most of the machine learning applications that use the Gromov-Wasserstein distance. However, when only distances between samples are available, the projection operation can not be carried anymore, while the computation of $GW$ is still possible. One can argue that it is possible to embed either isometrically those distances into a Hilbertian space, or at least with a low distortion, and then apply the presented technique. Our future line of work considers this option, as well as a possible direct reasoning on the distance matrix. For example, one should be able to consider geodesic paths (in a graph for instance) as the equivalent appropriate geometric object related to the line. This constitutes the direct follow-up of this work, as well as a better understanding of the accuracy of the estimated discrepancy with respect to the ambiant dimension and the projections number.

**Acknowledgements**

We would like to thank Nicolas Klutchnikoff for the hepful discussions. This work benefited from the support from OATMIL ANR-17-CE23-0012 project of the French National Research Agency (ANR). We gratefully acknowledge the support of NVIDIA Corporation with the donation of the Titan X GPU used for this research.

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
