[Supplementary Material]

# Supplementary material for paper
# Sliced Gromov-Wasserstein

**Notations** In the following $\mathcal{F}$ denotes the Fourrier transform. For a probability measure $\mu \in \mathcal{P}(\mathbb{R}^p)$ and for $s \in \mathbb{R}^p$, it is defined by $\mathcal{F}_\mu(s) = \int e^{-2i\pi\langle s,x\rangle} d\mu(x)$.

## 1 Proof for the QAP

In this section we aim at proving the new special case of the QAP, which is recalled in the next theorem:

**Theorem 1.1.** *A new special case of the QAP. For real numbers $x_1 \leq ... \leq x_n$ and $y_1 \leq ... \leq y_n$ then*

$$\min_{\sigma \in S_n} \sum_{i,j} \left( (x_i - x_j)^2 - (y_{\sigma(i)} - y_{\sigma(j)})^2 \right)^2 \tag{1}$$

*is achieved either by the identity permutation $\sigma(i) = i$ or the anti-identity permutation $\sigma(i) = n + 1 - i$.*

*Proof.* Let us note $\mathcal{I} = \{x, y \in \mathbb{R}^n \times \mathbb{R}^n | x_1 \leq x_2 \leq \cdots \leq x_n , y_1 \leq y_2 \leq \cdots \leq y_n\}$ and $S_n$ the set of all permutations of $\{1, ..., n\}$. We consider for $x, y \in \mathcal{I}$:

$$\max_{\sigma \in S_n} Z(x, y, \sigma) = \max_{\sigma \in S_n} \sum_{i,j} (x_i - x_j)^2 (y_{\sigma(i)} - y_{\sigma(j)})^2 \tag{2}$$

The original problem is equivalent to maximizing $Z(x, y, \sigma)$ over $S_n$. For any $x, y \in \mathcal{I}$, we recall the rearrangement inequality:

$$\forall \sigma \in S_n, \ \sum_i x_i y_{n+1-i} \leq \sum_i x_i y_{\sigma(i)} \leq \sum_i x_i y_i \tag{3}$$

We will prove that it suffices to solve a problem of the form $\underset{\sigma \in S_n}{\mathrm{argmax}} \left( \sum_i x_i y_{\sigma(i)} \right)^2$ in order to recover the optimal solution.

Now, given $x, y \in \mathcal{I}$, we define $X \overset{\text{def}}{=} \sum_i x_i$ and $Y \overset{\text{def}}{=} \sum_i y_i$. Then:

$$\max_{\sigma \in S_n} Z(x, y, \sigma) = \max_{\sigma \in S_n} \sum_{i,j} (x_i - x_j)^2 (y_{\sigma(i)} - y_{\sigma(j)})^2$$

$$= \max_{\sigma \in S_n} \sum_{i,j} (x_i^2 + x_j^2)(y_{\sigma(i)}^2 + y_{\sigma(j)}^2) - 2 \sum_{i,j} x_i x_j (y_{\sigma(i)}^2 + y_{\sigma(j)}^2) - 2 \sum_{i,j} y_{\sigma(i)} y_{\sigma(j)} (x_i^2 + x_j^2)$$

$$+ 4 \sum_{i,j} x_i x_j y_{\sigma(i)} y_{\sigma(j)}$$

$$= \max_{\sigma \in S_n} 2n \sum_i x_i^2 y_{\sigma(i)}^2 - 2 \sum_{i,j} x_i x_j (y_{\sigma(i)}^2 + y_{\sigma(j)}^2) - 2 \sum_{i,j} y_{\sigma(i)} y_{\sigma(j)} (x_i^2 + x_j^2)$$

$$+ 4 \sum_{i,j} x_i x_j y_{\sigma(i)} y_{\sigma(j)} + 2(\sum_i x_i^2)(\sum_i y_i^2)$$

$$= \max_{\sigma \in S_n} 2n \sum_i x_i^2 y_{\sigma(i)}^2 - 4X \sum_i x_i y_{\sigma(i)}^2 - 4Y \sum_i x_i^2 y_{\sigma(i)} + 4 \sum_{i,j} x_i x_j y_{\sigma(i)} y_{\sigma(j)} + 2(\sum_i x_i^2)(\sum_i y_i^2)$$

$$\overset{(*)}{=} C + 2\Big( \max_{\sigma \in S_n} \sum_i n x_i^2 y_{\sigma(i)}^2 - 2 \sum_i (X x_i y_{\sigma(i)}^2 + Y x_i^2 y_{\sigma(i)}) + 2(\sum_i x_i y_{\sigma(i)})^2 \Big)$$

where in (*) we defined $C \overset{\text{def}}{=} 2(\sum_i x_i^2)(\sum_i y_i^2)$ the term that does not depend on $\sigma$. We define

$$W(x, y, \sigma) \overset{\text{def}}{=} \sum_i n x_i^2 y_{\sigma(i)}^2 - 2(X x_i y_{\sigma(i)}^2 + Y x_i^2 y_{\sigma(i)}) + 2(\sum_i x_i y_{\sigma(i)})^2$$

and

$$f(x_i, y_{\sigma(i)}) \overset{\text{def}}{=} n x_i^2 y_{\sigma(i)}^2 - 2(X x_i y_{\sigma(i)}^2 + Y x_i^2 y_{\sigma(i)}) = n x_i^2 y_{\sigma(i)}^2 - 2((\sum_i x_i) x_i y_{\sigma(i)}^2 + 4(\sum_i y_i) x_i^2 y_{\sigma(i)})$$

such that:

$$W(x, y, \sigma) = \sum_i f(x_i, y_{\sigma(i)}) + 2(\sum_i x_i y_{\sigma(i)})^2$$

With these new definitions we have proven:

$$\forall x, y \in \mathcal{I}, \ \operatorname*{argmax}_{\sigma \in S_n} Z(x, y, \sigma) = \operatorname*{argmax}_{\sigma \in S_n} W(x, y, \sigma) = \operatorname*{argmax}_{\sigma \in S_n} \sum_i f(x_i, y_{\sigma(i)}) + 2(\sum_i x_i y_{\sigma(i)})^2$$

$$\tag{4}$$

We also introduce for $x, y \in \mathcal{I}, \ b \in \mathbb{R}$:

$$g(x, y, b) \overset{def}{=} \sum_i f(x_i + b, y_{\sigma(i)})$$

which is a perturbated version of the cost by a constant $b$. Since we know that the original cost $Z(x, y, \sigma)$ is invariant by any translation of the points $x, y$ the idea is to find a constant $b^*$ such that $g(x, y, b^*) = 0$ to simplify the problem. We have:

$$g(x, y, b) = -(n\|x\|_2^2 + 2Y^2)b^2 - \big(4Y \sum_i [x_i y_{\sigma(i)}] + 2X\|x\|_2^2\big)b + \sum_i x_i y_{\sigma(i)}(n x_i y_{\sigma(i)} - 2X y_{\sigma(i)} - 2Y x_i)$$

with $\|x\|_2^2 = \sum_i x_i^2$. Indeed:

$$g(x, y, b) = \sum_i f(x_i + b, y_{\sigma(i)}) = \sum_i n(x_i + b)^2 y_{\sigma(i)}^2 - 2\big((X + nb)(x_i + b)y_{\sigma(i)}^2 + Y(x_i + b)^2 y_{\sigma(i)}\big)$$

$$= \sum_i n(x_i^2 + 2b x_i + b^2)y_{\sigma(i)}^2 - 2\big((X x_i + Xb + nb x_i + nb^2)y_{\sigma(i)}^2 + Y(x_i^2 + 2b x_i + b^2)y_{\sigma(i)}\big)$$

$$= \sum_i b^2 \big[n y_{\sigma(i)}^2 - 2n y_{\sigma(i)}^2 - 2Y y_{\sigma(i)}\big]$$

$$+ \sum_i b \big[2n x_i y_{\sigma(i)}^2 - 2X y_{\sigma(i)}^2 - 2n x_i y_{\sigma(i)}^2 - 4Y x_i y_{\sigma(i)}\big]$$

$$+ \sum_i \big[n x_i^2 y_{\sigma(i)}^2 - 2X x_i y_{\sigma(i)}^2 - 2Y x_i^2 y_{\sigma(i)}\big]$$

$$= -(n\|x\|_2^2 + 2Y^2)b^2 - \big(4Y \sum_i x_i y_{\sigma(i)} + 2X\|x\|_2^2\big)b + \sum_i x_i y_{\sigma(i)}(n x_i y_{\sigma(i)} - 2X y_{\sigma(i)} - 2Y x_i)$$

If $X, Y = 0$ then $g(x, y, b) = 0 \iff b = b^*(x, y, \sigma) = \frac{1}{\|x\|_2}\sqrt{\sum_i x_i^2 y_{\sigma(i)}^2}$.

In this way for $x, y \in \mathcal{I}$ with $X, Y = 0$ using (4):

$$
\begin{aligned}
W(x + b^*(x, y, \sigma)1_n, y, \sigma) &= g(x, y, b^*(x, y, \sigma)) + 2(\sum_i (x_i + b^*(x, y, \sigma))y_{\sigma(i)})^2 \\
&= 2(\sum_i (x_i y_{\sigma(i)} + b^*(x, y, \sigma)y_{\sigma(i)}))^2 \\
&= 2(\sum_i x_i y_{\sigma(i)} + b^*(x, y, \sigma)\sum_i y_{\sigma(i)})^2 \\
&= 2(\sum_i x_i y_{\sigma(i)} + b^*(x, y, \sigma)Y)^2 \\
&= 2(\sum_i x_i y_{\sigma(i)})^2
\end{aligned}
\tag{5}
$$

Moreover for $x, y \in \mathcal{I}$ we have by invariance of the cost *w.r.t.* any translation:

$$
\operatorname*{argmax}_{\sigma \in S_n} Z(x, y, \sigma) = \operatorname*{argmax}_{\sigma \in S_n} Z(x - \frac{1}{n}\sum_i x_i, y - \frac{1}{n}\sum_i y_i, \sigma)
$$
$$
= \operatorname*{argmax}_{\sigma \in S_n} Z(x', y', \sigma)
$$

with $x', y' \in \mathcal{I}$ and $\sum_i x_i' = \sum_i y_i' = 0$. So without loss of generality we can solve the original problem only for $x, y \in \mathcal{I}$ with $X, Y = 0$. In this case:

$$
\begin{aligned}
\operatorname*{argmax}_{\sigma \in S_n} Z(x, y, \sigma) &\overset{*}{=} \operatorname*{argmax}_{\sigma \in S_n} Z(x + b^*(x, y, \sigma)1_n, y, \sigma) \\
&\overset{**}{=} \operatorname*{argmax}_{\sigma \in S_n} W(x + b^*(x, y, \sigma)1_n, y, \sigma) \\
&\overset{***}{=} \operatorname*{argmax}_{\sigma \in S_n} (\sum_i x_i y_{\sigma(i)})^2
\end{aligned}
\tag{6}
$$

Where in (*) we used the translation invariance property of $Z$, in (**) we used (4) and in (***) we used (5)

Now let us discuss the term $(\sum_i x_i y_{\sigma(i)})^2$ with the rearrangement inequality (3):

- If $\sum_i x_i y_{n+1-i} \geq 0$, then everything is positive in (3) so that we have $(\sum_i x_i y_{\sigma(i)})^2 \leq (\sum_i x_i y_i)^2$ for any $\sigma \in S_n$. In this case the identity is the optimal permutation.

- If $\sum_i x_i y_i \leq 0$ then everything is negative in (3) so that we have $(\sum_i x_i y_{\sigma(i)})^2 \leq (\sum_i x_i y_{n+1-i})^2$. In this case the anti-identity is the optimal permutation.

- If $\sum_i x_i y_{n+1-i} < 0$ and $\sum_i x_i y_i > 0$ then using (3) again,

$$
(\sum_i x_i y_{\sigma(i)})^2 \leq \max\{(\sum_i x_i y_{n+1-i})^2, (\sum_i x_i y_i)^2)\}
$$

In this case the optimal permutation is achieved whether by the identity or the anti-identity permutation.

$\square$

## 2   Computing $GW$ in the 1d case

As seen in the previous theorem finding the optimal permutation $\sigma^*$ can be found in $O(n\log(n))$. Moreover the final cost can be written using binomial expansion:

$$
\begin{aligned}
\sum_{i,j}\left((x_i-x_j)^2-(y_{\sigma^*(i)}-y_{\sigma^*(j)})^2\right)^2 &= 2n\sum_i x_i^4 - 8\sum_i x_i^3\sum_i x_i + 6(\sum_i x_i^2)^2 \\
&\quad + 2n\sum_i y_i^4 - 8\sum_i y_i^3\sum_i y_i + 6(\sum_i y_i^2)^2 \\
&\quad - 4(\sum_i x_i)^2(\sum_i y_i)^2 \\
&\quad - 4n\sum_i x_i^2 y_{\sigma^*(i)}^2 + 8\sum_i((\sum_i x_i)x_i y_{\sigma^*(i)}^2 + (\sum_i y_i)x_i^2 y_{\sigma^*(i)}) \\
&\quad - 8(\sum_i x_i y_{\sigma^*(i)})^2
\end{aligned}
\tag{7}
$$

which can be computed in $O(n)$ operations.

## 3   Claims about $GW$

This section aims at proving some claims in the paper about $GW$. Let us recall the notations of the paper.

We consider discrete measures $\mu \in \mathcal{P}(\mathbb{R}^p)$ and $\nu \in \mathcal{P}(\mathbb{R}^q)$ with $p \leq q$ on euclidean spaces such that $\mu = \sum_{i=1}^n a_i \delta_{x_i}$ and $\nu = \sum_{i=1}^m b_j \delta_{y_j}$, where $a \in \Sigma_n$ and $b \in \Sigma_m$ are histograms.

Let $c_X : \mathbb{R}^p \times \mathbb{R}^p \mapsto \mathbb{R}_+$ (*resp.* $c_Y : \mathbb{R}^q \times \mathbb{R}^q \mapsto \mathbb{R}_+$) measuring the similarity between the points in $\mu$ (*resp.* $\nu$). The Gromov-Wasserstein ($GW$) distance is defined as:

$$
GW_2^2(c_X, c_Y, \mu, \nu) = \min_{\pi \in \Pi(a,b)} J(c_X, c_Y, \pi)
\tag{8}
$$

where

$$
J(c_X, c_Y, \pi) = \sum_{i,j,k,l}\left|c_X(x_i, x_k) - c_Y(y_j, y_l)\right|^2 \pi_{i,j}\pi_{k,l}
$$

### 3.1   $GW$ when squared euclidean distances are used

When $c_X, c_Y$ are distances it has been shown in [1] that $GW$ defines a distance on the space of metric measure spaces quotiented by the measure-preserving isometries. More precisely, $GW$ is symmetric, satisfies the triangle inequality and $GW_2^2(c_X, c_Y, \mu, \nu) = 0$ *iff* there exists $f : \operatorname{supp}(\mu) \to \operatorname{supp}(\nu)$ such that

$$
f \# \mu = \nu
\tag{9}
$$

$$
\forall x, x' \in \operatorname{supp}(\mu)^2, c_X(x, x') = c_Y(f(x), f(x'))
\tag{10}
$$

In the paper we claim the following lemma:

*Lemma* 3.1. Using previous notations with $c_X(x, x') = \|x - x'\|_{2,p}^2$, $c_Y(y, y') = \|y - y'\|_{2,q}^2$. Then $GW_2^2(c_X, c_Y, \mu, \nu) = 0$ *iff* there exists a measure preserving isometry from $\operatorname{supp}(\mu)$ to $\operatorname{supp}(\nu)$ which satisfies (9) and (10)

*Proof.* If such an function exists by considering the coupling $\pi = (I_d \times f)\#\mu$ it is clear that $\pi$ is optimal and has a null cost so that $GW_2^2(c_X, c_Y, \mu, \nu) = 0$. Conversely, $GW_2^2(c_X, c_Y, \mu, \nu) = 0$

implies that $c_X(x, x') = c_Y(y, y')$ for all $(x, y), (x', y')$ in the support of an optimal plan $\pi^*$. This suffices to prove the existence of a measure preserving isometry (see (a) in Proof of Theorem 5.1 in [1]) □

## 3.2 Equivalence between $GM$ and $GW$ in the discrete case

This paragraph aims at proving the equivalence between $GM$ and $GW$. We will prove the following theorem (that is more general than the one used in the paper which only considers one-dimensional measures):

**Theorem 3.2.** *Equivalence between $GW$ and $GM$ for discrete measures*

*With $\mu$, $\nu$ defined previously and $c_X(x, x') = \|x - x'\|_{2,p}^2$, $c_Y(y, y') = \|y - y'\|_{2,q}^2$. Let us suppose also that $m = n$ and $\forall i \in \{1, ..., n\}, a_i = b_i = \frac{1}{n}$*

*Then $GW_2(c_X, c_Y, \mu, \nu) = GM_2(c_X, c_Y, \mu, \nu)$.*

*Proof.* The proof is essentially based on theoretical results from [2]. This paper considers the following energy minimizing problem:

$$\min_{X \in \mathcal{F}} E(X) \tag{11}$$

where $\mathcal{F} \subset \mathbb{R}^{n \times n}$ is a collection of matchings between the vertices of two graphs. More precisely the paper focuses on $E(X)$ of the form $E(X) = -\text{tr}(BX^TAX)$ and $\mathcal{F} = S_n$ the set of all permutations of $\{1, ..., n\}$. In fact, the $GM$ problem defined in the paper is equivalent to (11) by considering $A_{ij} = \|x_i - x_j\|_{2,p}^2$ and $B_{ij} = \|y_i - y_j\|_{2,q}^2$

Authors consider the set of doubly stochastic matrices (which is the convex-hull of $S_n$):

$$DS = \{X \in \mathbb{R}^{n \times n} \text{ s.t. } X1 = X^T1 = 1, X \geq 0\}$$

Minimizing $E(X)$ over $DS$ is equivalent to solving the $GW$ distance when $a_i = b_j = \frac{1}{n}$. The paper claims that if $E(X)$ is a *conditionally concave energy* then $\min_{X \in S_n} E(X)$ and $\min_{X \in DS} E(X)$ coincide.

This is verified when both $A$ and $B$ are conditionally positive (or negative) definite of order 1 (Theo 1 in [2]). Yet $A$ and $B$ defined previously satisfy this property (see examples under Definition 2 in [2]) and so $GW$ and $GM$ coincide.

□

## 4 Properties of SGW

$\|.\|$ is a norm on $\mathbb{R}^p$. To state the properties of $SGW$, we will need the Arzela-Ascoli Theorem. Let $(X, d)$ be a compact metric space and $C(X, \mathbb{R}^p)$ the space of all continuous functions from $X$ to $\mathbb{R}^p$. We recall:

- A family $\mathcal{F} \subset C(X, \mathbb{R}^p)$ is *bounded* means that there exists a positive constant $M < \infty$ such that $\|f(x)\| \leq M$ for all $x \in X$ and $f \in \mathcal{F}$
- A family $\mathcal{F} \subset C(X, \mathbb{R}^p)$ is *equicontinuous* means that for every $\epsilon > 0$ there exists $\delta > 0$ (which depends only on $\epsilon$) such that for $x, y \in X$:

$$d(x, y) < \delta \Rightarrow \|f(x) - f(y)\| < \epsilon \quad \forall f \in \mathcal{F}.$$

The Arzela-Ascoli Theorem states that if $(f_n)_{n \in \mathbb{N}}$ is a sequence in $C(X, \mathbb{R}^p)$ that is bounded and equicontinuous then it has a uniformly convergent subsequence.

We recall the theorem (measures $\mu$ and $\nu$ are defined discrete measures with the same number of atoms):

**Theorem 4.1.** *Properties of $SGW$*

- *For all $\Delta$, $SGW_\Delta$ and $RISGW$ are translation invariant. $RISGW$ is also rotational invariant when $p = q$, more precisely if $Q \in \mathcal{O}(p)$ is an orthogonal matrix, $RISGW(Q\#\mu, \nu) = RISGW(\mu, \nu)$*

- *$SGW$ and $RISGW$ are pseudo-distances on $\mathcal{P}(\mathbb{R}^p)$, i.e they are symetric, satisfy the triangle inequality and $SGW(\mu, \mu) = RISGW(\mu, \mu) = 0$.*

- *For $\mu, \nu \in \mathcal{P}(\mathbb{R}^p) \times \mathcal{P}(\mathbb{R}^p)$, if $SGW(\mu, \nu) = 0$ then $\mu$ and $\nu$ are isomorphic for the distance induced by the $\ell_1$ norm on $\mathbb{R}^p$. In particular this implies $GW_2(d_{\|.\|_{1,p}}, \mu, \nu) = 0$.*

The invariance by translation is clear since the costs are invariant by translation of the support of the measures. The pseudo-distances properties are straightforward thanks to the properties of $GW$.

**Theorem 4.2.** *For $\mu, \nu \in \mathcal{P}(\mathbb{R}^p) \times \mathcal{P}(\mathbb{R}^p)$, if $SGW(\mu, \nu) = 0$ then $\mu$ and $\nu$ are isomorphic for the distance induced by the $\ell_1$ norm on $\mathbb{R}^p$. In particular this implies that $GW_2(d_{\|.\|_{1,p}}, \mu, \nu) = 0$.*

*Proof.* In the proof $\|.\|$ denotes the $\ell_1$ norm and $\|.\|_2$ denotes the $\ell_2$ norm. We note $M_\mu = \max_{x \in \text{supp}(\mu)} \|x\|_2$ and $M_\nu = \max_{y \in \text{supp}(\nu)} \|y\|_2$. The objective is to prove that if $SGW(\mu, \nu) = 0$ there exists a surjective function $f : \text{supp}(\mu) \to \text{supp}(\nu)$ such that $f$ is an isometry for the $\ell_1$ norm $(\forall x, x' \in \text{supp}(\mu), \|f(x) - f(x')\| = \|x - x'\|)$ and pushes $\mu$ into $\nu$ ($f\#\mu = \nu$).

The proof is divided into four parts. In the first one, we construct an "almost orthogonal" basis on which measures are isomorphic. Building upon this result we define a sequence of functions from $\text{supp}(\mu)$ to $\text{supp}(\nu)$ and show that it has a convergent subsequence. We conclude by proving that the limit of the subsequence is actually a good candidate for being the isometry we are looking for.

**There exists an "almost orthogonal" basis on which measures are isomorphic** Suppose that $SGW(\mu, \nu) = 0$. Then by the Gromov-Wasserstein properties for almost all $\theta \in \mathbf{S}^{p-1}$:

$$
\begin{aligned}
&\exists T_\theta : \mathbb{R} \mapsto \mathbb{R}, \text{ surjective s.t. } T_\theta\#(P_\theta\#\mu) = P_\theta\#\nu \\
&\forall x, x' \in \text{supp}(P_\theta\#\mu), |T_\theta(x) - T_\theta(x')| = |x - x'|
\end{aligned}
\qquad (\mathcal{Q}_\theta)
$$

We want to construct a basis $(e_1, ..., e_p)$ as orthogonal as possible such that for all $i$ we have $\mathcal{Q}_{e_i}$. In order to do so, we consider for $n \in \mathbb{N}^*$,

$$
\mathcal{B}_p^n = \{(e_1, ..., e_p) \in (\mathbf{S}^{p-1})^p \text{ s.t. } |\langle e_i, e_j \rangle| < \frac{1}{n}\}
$$

and

$$
Q = \{(e_1, ..., e_p) \in (\mathbf{S}^{p-1})^p \text{ s.t. } \forall i \in \{1, ..., p\}, \mathcal{Q}_{e_i}\}
$$

We also note $\lambda_{p-1}^{\otimes p}$ the product measure $\lambda_{p-1} \otimes ... \otimes \lambda_{p-1}$. $\mathcal{B}_p^n$ is an open set as inverse image by a continuous function of an open set. Then $\lambda_{p-1}^{\otimes p}(\mathcal{B}_p^n) > 0$. Moreover, since for almost all $\theta \in \mathbf{S}^{p-1}$ we have $\mathcal{Q}_\theta$ then $\lambda_{p-1}^{\otimes p}(Q) > 0$ and so $\lambda_{p-1}^{\otimes p}(\mathcal{B}_p^n \cap Q) > 0$.

In this way we can consider $(e_1(n), ..., e_p(n)) \in \mathcal{B}_p^n \cap Q$. If $n > p - 1$ the Gram matrix of $(e_1(n), ..., e_p(n))$ is strictly diagonal dominant, thus invertible, such that $(e_1(n), ..., e_p(n))$ is a basis. In the following $n > p - 1$ and $(e_1(n), ..., e_p(n))$ is the basis constructed with the previous procedure. The idea is to construct the isometry thanks to this "almost" orthogonal basis.

In the proof $x_i$ denotes the $i$th coordinate of $x$ in the standard euclidean basis. For $x \in \mathbb{R}^p$, we can write in the new basis:

$$
x = \sum_{i=1}^{p} [\langle x, e_i(n) \rangle + R(x, e_i(n))]e_i
$$

with $R(x, e_i(n)) \stackrel{\text{def}}{=} x_i - \langle x, e_i(n) \rangle$ and $|R(x, e_i(n))| = o(\frac{1}{n})$.

Indeed,

$$x = \sum_{i=1}^{p} x_i e_i \implies \text{for j } \langle x, e_j \rangle = \sum_{i=1}^{p} x_i \langle e_i, e_j \rangle$$

$$\implies x_j - \langle x, e_j \rangle = \sum_{i \neq j} x_i \langle e_i, e_j \rangle$$

$$\implies |R(x, e_j(n))| = |\sum_{i \neq j} x_i \langle e_i, e_j \rangle|$$

$$\implies |R(x, e_j(n))| \leq \frac{1}{n} \sum_{i \neq j} |x_i| \leq \frac{C_{p,\mu}}{n}$$

with some constant $C_{p,\mu}$ that only depends on $\mu$ and $p$ (it is actually in the form $C * M_\mu$ since all norms are equivalent). Also in the same way for $s, y \in \mathbb{R}^p \times \mathbb{R}^p$ we can rewrite their scalar product:

$$\langle s, y \rangle = \sum_{i=1}^{p} \langle s, e_i(n) \rangle \langle y, e_i(n) \rangle + \tilde{R}(s, y) \tag{12}$$

with:

$$\tilde{R}(s, y) \overset{\text{def}}{=} \langle s, y \rangle - \sum_{i=1}^{p} \langle s, e_i(n) \rangle \langle y, e_i(n) \rangle = \sum_{i \neq j} \langle s, e_i(n) \rangle \langle y, e_i(n) \rangle \langle e_j(n), e_i(n) \rangle$$

$$+ \sum_{i,j} \langle s, e_i(n) \rangle R(y, e_j(n)) \langle e_j(n), e_i(n) \rangle$$

$$+ \sum_{i,j} \langle y, e_j(n) \rangle R(s, e_i(n)) \langle e_j(n), e_i(n) \rangle$$

$$+ \sum_{i,j} R(y, e_j(n)) R(s, e_i(n)) \langle e_j(n), e_i(n) \rangle$$

and with the same calculus than for $R$ we have $|\tilde{R}(s, y)| = o(\frac{1}{n})$.

**Construction of a "good" sequence**    Using previous notations we define:

$$\forall n > p - 1, \ \forall x \in \text{supp}(\mu), \ f_n(x) = (T_{e_1(n)}(\langle x, e_1(n) \rangle), ..., T_{e_p(n)}(\langle x, e_p(n) \rangle)) \tag{13}$$

Clearly all $f_n$ are surjectives and continuous since all $T_{e_k(n)}$ are, thanks to $\mathcal{Q}_{e_k(n)}$. We will show that we can derive from $(f_n)_{n \in \mathbb{N}}$ a good candidate for being the isometry we are looking for. The sequence satisfies the following properties:

*Lemma* 4.3.  Properties of $(f_n)_{n \in \mathbb{N}}$

$$\forall n \in \mathbb{N}, \forall x, x' \in \text{supp}(\mu)^2, \ \left| \|f_n(x) - f_n(x')\| - \|x - x'\| \right| = o(\frac{1}{n}) \tag{14}$$

$$\forall s \in \mathbb{R}^p, \ |\mathcal{F}_{f_n \# \mu}(s) - \mathcal{F}_\nu(s)| = o(\frac{1}{n}) \tag{15}$$

For clarity purposes, we prove this lemma at the end of the proof. In the next paragraph we will show that we can extract a convergent subsequence from $(f_n)_{n \in \mathbb{N}}$ thanks to Arzela-Ascoli Theorem.

**We can extract from $(f_n)_{n \in \mathbb{N}}$ a convergent subsequence**    We will show that $(f_n)_{n \in \mathbb{N}}$ is equicontinuous. Let $\epsilon > 0$, using (14) there exists a $N \in \mathbb{N}$ such that we have for all $x, x' \in \text{supp}(\mu)$:

$$\|f_n(x) - f_n(x')\| \leq \epsilon + \|x - x'\| \ \text{ for all } n \geq N$$

Now let $\delta < \epsilon$. Suppose that $\|x - x'\| < \delta$ then

$$\|f_n(x) - f_n(x')\| < \epsilon + \delta < 2\epsilon \ \text{ for all } n \geq N$$

Without loss of generality we can reindex $(f_n)_{n \in \mathbb{N}}$ for $n$ large enough ($n \geq N$) so that $(f_n)_{n \in \mathbb{N}}$ is equicontinuous with the previous argument.

Moreover $(f_n)_{n \in \mathbb{N}}$ is also bounded. Indeed for all $n \in \mathbb{N}$ since $T_{e_k(n)}$ is a surjective isometry from $\mathrm{supp}(P_{e_k(n)}\#\mu)$ to $\mathrm{supp}(P_{e_k(n)}\#\nu)$ then it is necessarily a bijection. So for all $x \in \mathrm{supp}(\mu)$ there exists a $y_0(x, n) \in \mathrm{supp}(\nu)$ such that $T_{e_k(n)}(\langle x, e_k(n)\rangle) = \langle y_0(x, n), e_k(n)\rangle$. In this way $|T_{e_k(n)}(\langle x, e_k(n)\rangle)| = |\langle y_0(x, n), e_k(n)\rangle| \leq \|y_0(x, n)\|_2 \leq M_\nu$ by Cauchy-Swartz.

So we have for $n \in \mathbb{N}$, $x \in \mathrm{supp}(\mu)$,

$$\|f_n(x)\|_2^2 = \sum_{k=1}^{p} |T_{e_k(n)}(\langle x, e_k(n)\rangle)|^2 \leq p M_\nu$$

Since on $\mathbb{R}^p$ all norms are equivalent it is sufficient to state the existence of a constant $C$ such that $\forall x \in \mathbb{R}^p, n \in \mathbb{N}, \|f_n(x)\| \leq C$.

To summarize $(f_n)_{n \in \mathbb{N}}$ is a bounded and equicontinuous sequence so by Arzela-Ascoli Theorem $(f_n)_{n \in \mathbb{N}}$ has a uniformly convergent subsequence: $f_{\phi(n)} \underset{n \to \infty}{\overset{u}{\to}} f$

Moreover eq. (4) states that for all $s \in \mathbb{R}^p$, $\mathcal{F}_{f_n\#\mu}(s) \underset{n \to \infty}{\to} \mathcal{F}_\nu(s)$. In this way $(\mathcal{F}_{f_n\#\mu}(s))_{n \in \mathbb{N}}$ is a convergent real valued sequence, so every adherence value goes to the same limit, hence $\mathcal{F}_{f_{\phi(n)}\#\mu}(s) \underset{n \to \infty}{\to} \mathcal{F}_\nu(s)$.

**The function $f$ is a measure preserving isometry from $\mathrm{supp}(\mu)$ to $\mathrm{supp}(\nu)$** Let $\epsilon_1 > 0, s \in \mathbb{R}^p$, there exists from previous statements $N_0, N_1 \in \mathbb{N}$ such that for $n \geq N_0$, $|\mathcal{F}_{f_{\phi(n)}\#\mu}(s) - \mathcal{F}_\nu(s)| < \epsilon_1$ and $n \geq N_1$, $|\mathcal{F}_{f_{\phi(n)}\#\mu}(s) - \mathcal{F}_{f\#\mu}(s)| < \epsilon_1$.

Let $n \geq \max(N_0, N_1)$

$$|\mathcal{F}_{f\#\mu}(s) - \mathcal{F}_\nu(s)| \leq |\mathcal{F}_{f_{\phi(n)}\#\mu}(s) - \mathcal{F}_\nu(s)| + |\mathcal{F}_{f_{\phi(n)}\#\mu}(s) - \mathcal{F}_{f\#\mu}(s)|$$
$$< 2\epsilon_1$$

As this result holds true for any $\epsilon_1 > 0$ we have $\mathcal{F}_{f\#\mu}(s) = \mathcal{F}_\nu(s)$ and by injectivity of the Fourrier transform $f\#\mu = \nu$ such that $f$ is measure preserving.

In the same way for any $x, x' \in \mathrm{supp}(\mu), \epsilon_2 > 0$ and $n$ large enough

$$\left| \|f(x) - f(x')\| - \|x - x'\| \right| \leq \left| \|f_{\phi(n)}(x) - f_{\phi(n)}(x')\| - \|f(x) - f(x')\| \right|$$
$$+ \left| \|f_{\phi(n)}(x) - f_{\phi(n)}(x')\| - \|x - x'\| \right|$$
$$< 2\epsilon_2$$

using $f_{\phi(n)} \underset{n \to \infty}{\overset{u}{\to}} f$ and (14). As this result holds true for any $\epsilon_2 > 0$ we have $\|f(x) - f(x')\| = \|x - x'\|$ for any $x, x' \in \mathrm{supp}(\mu)$.

To conclude $f$ is a surjective isometry that preserves the measures so $\mu$ and $\nu$ are isomorphic. By the properties of $GW$ the Gromov-Wasserstein distance defined previously also vanishes.

$\square$

In the previous proof we admitted the lemma 4.3 that we now prove:

*Proof.* Proof of Lemma 4.3

We have to show that:

$$\forall n \in \mathbb{N}, \forall x, x' \in \mathrm{supp}(\mu)^2, \left| \|f_n(x) - f_n(x')\| - \|x - x'\| \right| = o(\frac{1}{n})$$

$$\forall s \in \mathbb{R}^p, \ |\mathcal{F}_{f_n \# \mu}(s) - \mathcal{F}_\nu(s)| = o(\frac{1}{n})$$

For $x, x' \in \text{supp}(\mu)$:

$$\|f_n(x) - f_n(x')\| = \sum_{k=1}^{p} |T_{e_k(n)}(\langle x, e_k(n)\rangle) - T_{e_k(n)}(\langle x', e_k(n)\rangle)|$$

$$\overset{(*)}{=} \sum_{k=1}^{p} |\langle x, e_k(n)\rangle - \langle x', e_k(n)\rangle|$$

$$= \sum_{k=1}^{p} |\langle x - x', e_k\rangle|$$

where in (*) we used $\mathcal{Q}_{e_k(n)}$ since $\langle x, e_k(n)\rangle \in \text{supp}(P_{e_k(n)}\#\mu)$ (idem for $x'$). In this way:

$$\big|\|f_n(x) - f_n(x')\| - \|x - x'\|\big| = \big|\sum_{k=1}^{p} |\langle x - x', e_k(n)\rangle| - |x_k - x'_k|\big|$$

$$\leq \sum_{k=1}^{p} \big||\langle x - x', e_k(n)\rangle| - |x_k - x'_k|\big|$$

$$\leq \sum_{k=1}^{p} |\langle x - x', e_k(n)\rangle - (x_k - x'_k)|$$

$$= \sum_{k=1}^{p} |R(x - x', e_k(n))| = o(\frac{1}{n})$$

Hence

$$\big|\|f_n(x) - f_n(x')\| - \|x - x'\|\big| = o(\frac{1}{n}) \tag{16}$$

Moreover we have by definition of the Fourrier transform, for $s \in \mathbb{R}^P$,

$$\mathcal{F}_{f_n \# \mu}(s) = \int e^{-2i\pi\langle s, f_n(x)\rangle} d\mu(x)$$

$$= \int e^{-2i\pi \sum_{k=1}^{p} s_k T_{e_k(n)}(\langle x, e_k(n)\rangle)} d\mu(x) \tag{17}$$

$$= \prod_{k=1}^{p} \int e^{-2i\pi s_k T_{e_k(n)}(\langle x, e_k(n)\rangle)} d\mu(x)$$

Then using $(\mathcal{Q}_\theta)$ we have for all $k \in \{1, ..., p\}$, and any real $t \in \mathbb{R}$

$$\mathcal{F}_{T_{e_k(n)}\#(P_{e_k(n)}\#\mu)}(t) = \mathcal{F}_{P_{e_k(n)}\#\nu}(t)$$

$$\implies \int e^{-2i\pi t T_{e_k(n)}(\langle e_k(n), x\rangle)} d\mu(x) = \int e^{-2i\pi t \langle e_k(n), y\rangle} d\nu(y)$$

So by applying this results for $t = s_k$ we have:

$$\int e^{-2i\pi s_k T_{e_k(n)}(\langle x, e_k(n)\rangle)} d\mu(x) = \int e^{-2i\pi s_k \langle e_k(n), y\rangle} d\nu(y) \tag{18}$$

Combining (18) and (17):

$$\mathcal{F}_{f_n \# \mu}(s) = \prod_{k=1}^{p} \int e^{-2i\pi s_k \langle e_k(n), y\rangle} d\nu(y) \tag{19}$$

So:

$$|\mathcal{F}_{f_n\#\mu}(s) - \mathcal{F}_\nu(s)| = |\int e^{-2i\pi\langle s, f_n(x)\rangle}d\mu(x) - \int e^{-2i\pi\langle s,y\rangle}d\nu(y)|$$

$$\overset{*}{=} |\int e^{-2i\pi\langle s, f_n(x)\rangle}d\mu(x) - \int e^{-2i\pi[\sum_{k=1}^p \langle s,e_k(n)\rangle\langle e_k(n),y\rangle + \tilde{R}(s,y)]}d\nu(y)|$$

$$\overset{**}{=} |\prod_{k=1}^p \int e^{-2i\pi s_k\langle e_k(n),y\rangle}d\nu(y) - \int e^{-2i\pi\tilde{R}(s,y)}e^{-2i\pi\sum_{k=1}^p \langle s,e_k(n)\rangle\langle e_k(n),y\rangle}d\nu(y)|$$

$$= |\int e^{-2i\pi\sum_{k=1}^p s_k\langle e_k(n),y\rangle}d\nu(y) - \int e^{-2i\pi\tilde{R}(s,y)}e^{-2i\pi\sum_{k=1}^p \langle s,e_k(n)\rangle\langle e_k(n),y\rangle}d\nu(y)|$$

$$\overset{***}{=} |\int e^{-2i\pi\sum_{k=1}^p (\langle s,e_k(n)\rangle + R(s,e_k(n)))\langle e_k(n),y\rangle}d\nu(y)$$

$$- \int e^{-2i\pi\tilde{R}(s,y)}e^{-2i\pi\sum_{k=1}^p \langle s,e_k(n)\rangle\langle e_k(n),y\rangle}d\nu(y)|$$

$$= |\int e^{-2i\pi\sum_{k=1}^p R(s,e_k(n))\langle e_k(n),y\rangle}e^{-2i\pi\sum_{k=1}^p \langle s,e_k(n)\rangle\langle e_k(n),y\rangle}d\nu(y)$$

$$- \int e^{-2i\pi\tilde{R}(s,y)}e^{-2i\pi\sum_{k=1}^p \langle s,e_k(n)\rangle\langle e_k(n),y\rangle}d\nu(y)|$$

$$= |\int e^{-2i\pi\sum_{k=1}^p \langle s,e_k(n)\rangle\langle e_k(n),y\rangle}\left(e^{-2i\pi\sum_{k=1}^p R(s,e_k(n))\langle e_k(n),y\rangle} - e^{-2i\pi\tilde{R}(s,y)}\right)d\nu(y)|$$

$$\leq \int |e^{-2i\pi\sum_{k=1}^p R(s,e_k(n))\langle e_k(n),y\rangle} - e^{-2i\pi\tilde{R}(s,y)}|d\nu(y)$$

$$= \int |e^{-2i\pi\tilde{R}(s,y)}\left(e^{-2i\pi(\sum_{k=1}^p R(s,e_k(n))\langle e_k(n),y\rangle - \tilde{R}(s,y))} - 1\right)|d\nu(y)$$

$$\leq \int |e^{-2i\pi(\sum_{k=1}^p R(s,e_k(n))\langle e_k(n),y\rangle - \tilde{R}(s,y))} - 1|d\nu(y)$$

$$= \int |2ie^{-i\pi(\sum_{k=1}^p R(s,e_k(n))\langle e_k(n),y\rangle - \tilde{R}(s,y))}\sin(\pi(\sum_{k=1}^p R(s,e_k(n))\langle e_k(n),y\rangle - \tilde{R}(s,y))|d\nu(y)$$

$$\leq \int |\sin(\pi(\sum_{k=1}^p R(s,e_k(n))\langle e_k(n),y\rangle - \tilde{R}(s,y))|d\nu(y)$$

$$\leq \pi\int (\sum_{k=1}^p |R(s,e_k(n))\langle e_k(n),y\rangle| + |\tilde{R}(s,y)|)d\nu(y)$$

$$\overset{****}{=} o(\frac{1}{n})$$

where in (*) we used the expression in the new base of the scalar product $\langle s,y\rangle$, in (**) we used (19), in (***) the expression of $s_k$ w.r.t the new base and in (****) the fact that each term is $o(\frac{1}{n})$ In this way:

$$|\mathcal{F}_{f_n\#\mu}(s) - \mathcal{F}_\nu(s)| = o(\frac{1}{n}) \tag{20}$$

□

For the invariance by rotation if $p = q$ then $\mathbb{V}_p(\mathbb{R}^p)$ is bjective with $\mathcal{O}(p)$ so for $Q \in \mathcal{O}(p)$:

$$
\begin{aligned}
RISGW(Q\#\mu, \nu) &= \min_{\Delta \in \mathbb{V}_p(\mathbb{R}^p)} SGW_\Delta(Q\#\mu, \nu) \\
&= \min_{\Delta \in \mathcal{O}(p)} SGW_\Delta(Q\#\mu, \nu) \\
&= \min_{\Delta \in \mathcal{O}(p)} \mathbb{E}_{\theta \sim \lambda_{q-1}} [GW(d^2, P_\theta\#(\Delta Q\#\mu), P_\theta\#\nu)] \quad (21) \\
&= \min_{\Delta' \in \mathcal{O}(p)} \mathbb{E}_{\theta \sim \lambda_{q-1}} [GW(d^2, P_\theta\#\Delta'\#\mu, P_\theta\#\nu)] \\
&= RISGW(\mu, \nu)
\end{aligned}
$$

On the other side for $\nu$ a change of formula on theta gives the result.

# 5   Algorithm for $SGW$

---

Sliced Gromov-Wasserstein for discrete measures

1: $p < q$, $\mu = \frac{1}{n} \sum_{i=1}^n \delta_{x_i} \in \mathcal{P}(\mathbb{R}^p)$ and $\nu = \frac{1}{n} \sum_{i=1}^n \delta_{y_j} \in \mathcal{P}(\mathbb{R}^q)$
2: $\forall i, x_i \leftarrow \Delta(x_i)$, sample uniformly $(\theta_l)_{l=1,\dots,L} \in \mathbf{S}^{q-1}$
3: **for** $l = 1, \dots, L$ **do**
4:    Sort $(\langle x_i, \theta_l \rangle)_i$ and $(\langle y_j, \theta_l \rangle)_j$ in increasing order
5:    Solve (1) for reals $(\langle x_i, \theta_l \rangle)_i$ and $(\langle y_j, \theta_l \rangle)_j$, $\sigma_{\theta_l}$ is the solution ($\sigma_{\theta_l} \in$ Anti-Id or Id )
6: **end for**
7: return $\frac{1}{n^2 L} \sum_{l=1}^L \sum_{i,k=1}^n \left( \langle x_i - x_k, \theta_l \rangle^2 - \langle y_{\sigma_{\theta_l}(i)} - y_{\sigma_{\theta_l}(k)}, \theta_l \rangle^2 \right)^2$

---

In practice, the computation trick presented in Equation (7) can be used to make the complexity of the computation in line 7 linear with $n$.

# 6   $SW_\Delta$ and $RISW$

Analogously to $SGW$ we can define for the Sliced-Wasserstein distance $SW_\Delta(\mu, \nu)$ for $\mu, \nu \in \mathcal{P}(\mathbb{R}^p) \times \mathcal{P}(\mathbb{R}^q)$ with $p \neq q$ and its rotational invariant counterpart as:

$$
\begin{aligned}
SW_\Delta(\mu, \nu) &= \fint_{\mathbf{S}^{q-1}} SW(P_\theta\#\mu_\Delta, P_\theta\#\nu) d\theta \\
RISW(\mu, \nu) &= \min_{\Delta \in \mathbb{V}_q(\mathbb{R}^p)} SW_\Delta(\mu, \nu)
\end{aligned} \quad (22)
$$

where $SW$ is the Sliced-Wasserstein distance. The complexity for computing $SW_\Delta$ is $O(Ln(p + q + \log(n)))$ which is exactly the same complexity as $SGW_\Delta$. With these formulations, we can perform the same experiment as for RISGW on the spiral dataset. The optimisation on the Stiefel manifold is performed using Pymanopt as for $SGW$. Results are reported in Figure 1. As one can see, $RISW$ is rotational invariant on average whereas $SW$ is not. One can also note that, due to the sampling process of the spiral dataset, the variance is quite large. This can be explained by the fact that, unlike $SGW$, the Sliced-Wasserstein may realign the distributions without taking the rotation into account.

# 7   Supplementary results for the $SGW$ GAN Section

We give here supplementary results for the $SGW$ GAN experiment in Fig. 2, where we consider first a generator that outputs 2D samples, with a two dimensional target, and then a generator that generates 3D samples form a 2D target distribution. Here again, the results are reported for 1000 epochs.

Figure 1: Illustration of $SW$, $RISW$ on spiral datasets for varying rotations on discrete 2D spiral datasets. (left) Examples of spiral distributions for source and target with different rotations. (right) Average value of $SW$ and $RISW$ with $L = 20$ as a function of rotation angle of the target. Colored areas correspond to the 20% and 80% percentiles.

Figure 2: Using $SGW$ in a GAN loss. The three rows depicts three different examples. First row is 2D (Generator) to 2D (Target) , Second 3D to 2D. First column is initialization, second one is at 100 Epochs, third one at 1000. Last column depicts the target distribution.