[Reviews · NeurIPS 2019]

Reviewer 1



### AFTER REBUTTAL#### I thank the authors for addressing my concerns, I change my score to 7. ################################################################# This paper presents an interesting extension of the slicing trick use to compute the Wasserstein distance to the Gromov-Wasserstein distance. To do so, they provide a closed form solution to GW in 1D. The authors claim a huge gain in computational time with their method compared to previous methods for GW. The paper is overall clear and well written, but I seem to be missing some key issues : - in the proof for QAP in 1D, i don't understand how the problem is equivalent to (2) : how do you make the quadratic term with y_sigma vanish? - I am confused by the role of delta. I understand that it is meant to allow points to be projected onto theta, but then why does it appear in cases when p=q (in particular 1st point of theorem 3.3)? Regarding the experimental section: - I do not understant what you are computing in the spiral example? Simply GW between two spirals? if so, I don't see why GW would be used here, as both spirals are in 2D? It the idea is to get a rotation invariant metric, maybe it should be compared to other rotation invariant OT methods (Alvarez-Melis et al). - How does RISGW compare to other state of the art methods for GW in terms of CPU time? - The computational gain is impressive, but i am wondering what comes from SGW, and what comes from using PyKeops? I understand it might be too much to implement entropic GW with PyKeops but it would be interesting to see where you method stands with a simple pytorch implementation. - The setting that you use to compare runtimes is 2 measures in 2D, again for practical use it would make more sense to consider different distributions. More generally, I would like the experimental section to highlight : - the effect of the number of projection directions - the effect of the delta operator The paper is overall clear and well written, with very interesting ideas but for the moment the concerns that I have raised prevent me from giving it a clear accept.

Reviewer 2



after feedback: I am happy with the author feedback and maintain my score. === The article proposes a fast approximation of the Gromov Wasserstein distance, which allows comparing distributions defined possibly on different spaces, e.g. R^p and R^q for p different than q. The approximation draws on the sliced Wasserstein distance, and a new result on the quadratic assignment problem in one dimension. The method is novel, the new result on QAP in 1-dimensional is quite simple but certainly useful. The experiments are very convincing; Figure 4 is particularly fun. The article is a pleasant read. The method requires a choice of linear map Delta which could be elaborated on a bit more (what else apart from padding?). The rotational invariant version is appealing but also makes it clear that some of the appeal of the original GW distance is lost (namely the fact that no specific sets of invariances have to be specified for the original GW distance). The discussion section raises some interesting points, e.g. GW is sometimes used on spaces that are not Hilbertian, and projections on the real line might not always be an option; I will be curious to see how the authors will tackle such cases.

Reviewer 3



Post rebuttal: Thank you for your responses. First of all, I don't agree that the distances that I suggested are non-obvious, especially given the current content of the paper. They can be described even in one paragraph. And I am quite sure that, unless clarified either in the main document or the supplementary, this would be a common criticism that the paper would receive. Therefore, by trusting the authors that they will fix all the concerns (typos, computational concerns etc) **and** they will keep their promise to conduct experiments on the "baseline" distances, I am increasing my score to 7. =============================================== Originality: The paper considers the Gromov-Wasserstein distance and aims at proposing a new (pseudo) distance that has similar properties to GW but at the same time computationally less expensive. The authors use ideas from the sliced-Wasserstein distance, which requires projecting samples onto lines and an analytical solution for the 1d problem. The authors show that an analytical solution exists and they base their approach on this result. I think the overall approach is sufficiently original (of course the idea is based on many other existing ones, which is normal). Quality: The authors handle the problem with a mathematically rigorous way. The proposed approach is intuitive and logical. The experiments illustrate their claims. I have gone over the proofs. I have looked at the proof of Theorem 3.1 (of main paper), which I believe the main contribution of the paper (the new distance heavily depends on this result). I have looked at the proofs of the other results, but did not check line by line. Overall I did not spot a major issue. For the experiments: it should be nice if the authors added a plot that is similar to Fig 3, but contains the distance values on the y axis, to see the differences between these distances. For the proof of Thm3.1, I have the following comments: - Overall, I like the proof. It’s written quite clear and contains small clever tricks (I’m not sure if the authors are the first ones to come up with those but in any case I like the ideas). - Eqn 3 requires a reference. - Supp doc Line 16, I am not sure what $N$ is. I failed to find where it was defined. In any case it should be clearly defined in the proof. On the other hand, I have a more major concern about the proposed pseudo metric. The whole idea of using the GW distance is to be able to compare distributions supported on very different sets. The authors claim that this is the main advantage of GW over W (and SW obviously). But in eqn 4, they introduce the matrix \Delta, such that the point masses of \mu is first pushed forward by \Delta and hence they are brought to the same space as \nu, then they are projected by using \theta. Now, the question is, why can’t we do the same thing for the Wasserstein distance itself? One can very well define a new discrepancy as SW(\mu_#\Delta, \nu). In this case, being able to compare \mu and \nu with different domains is no longer an advantage over SW. In this respect, I strongly believe that the authors should define four new discrepancies: 1) W(\mu_#\Delta, \nu) 2) SW(\mu_#\Delta, \nu) 3) min_\Delta W(\mu_#\Delta, \nu) 4) min_\Delta SW(\mu_#\Delta, \nu) and compare these pseudo-metrics to theirs. I believe they can implement these very easily by using their existing code. Some of these discrepancies might have better computational properties but I don’t believe that it would degrade the value of the paper. Additional references: - Line 43: The authors can cite these two additional papers to be more complete: 1- Wu et al. “Sliced Wasserstein Generative Models”, CVPR 2019 2- Liutkus et al. ““Sliced-Wasserstein Flows: Nonparametric Generative Modeling via Optimal Transport and Diffusions”, ICML 2019 - Eqn 4, basically tells us that the point masses of \mu are projected by using \theta^T \Delta and the mass of \nu is projected by using \theta. In this respect, Eqn 5 (optimizing over \Delta), is in a way optimizing the projectors of \mu. Finding optimal projectors have been studied in the literature and the authors should cite the following papers around line 206. 1- Deshpande et al, “Max-Sliced Wasserstein Distance and its use for GANs”, CVPR 2019 2- Kolouri et al, “Generalized Sliced Wasserstein Distances”, arxiv 2019 3- Paty and Cuturi, “Subspace robust wasserstein distances”, ICML 2019 Clarity: The paper is overall clearly written and organized. I enjoyed reading it. However, I have some minor issues about some sentences. And there are many typos which need to be fixed. Details: - The title seems as if it is not finished. The authors should change it to something like “Sliced Gromov-Wasserstein Discrepancy” if it is still allowed. - I find the first sentence of the abstract a little bit awkward. The authors mention two distributions that do not lie in the same metric space. But if GW distance can be defined between these two distributions, they automatically lie in the metric space induced by GW, don’t they? The authors should clearly mention that they refer to the supports of the distributions. - The first sentence of the introduction is misleading. Defining distances over distributions is not the main goal of OT. - Line 26: This sentence implies that the Wasserstein distance can only be measured between two measures that are supported on the same set, which is not true. - Abstract, line 6, sentence starting with “among those”: this sentence creates the illusion that there is a family of Wasserstein distances and sliced-Wasserstein is one of them. The current theory cannot say W and SW are always equivalent (eg. measures supported on non-compact domains) - L188: SW and W haven’t been shown to be equivalent on general probability spaces. Bonnotte only showed that they are equivalent on compact domains. This should be clarified. - The references section contains “et al.”s. They should be replaced with full author names. Typos: - L8: propose -> proposes - L21: modelling -> modeling (The authors should stick to either British or American English, the text is mostly written in American English that’s why I’m suggesting this. These mistakes should be corrected in other parts of the paper as well). - L38: into -> onto - All instances of “monodimensional” to “unidimensional”. I haven’t seen the use of monodimensional before, then I have checked google, unidimensional is way more common. - L51: scenarii -> scenarios (I had to check google for this as well) - All instances of “euclidean” to “Euclidean” (there are many including the supp doc…) - L63: It’s not clear to me why the histograms are defined on the extended reals, a_i cannot be infinity anyways, why R* instead of R? - L69: the authors should add a \in \Sigma_n - L104: notations -> notation - L137: I suggest using other symbols for the matrices than a and b. They made me confused with the histograms that come after. - L154: resulting on -> resulting in - L180: the authors should mention that eqn 2 is the main reason why the computational complexity becomes O(n^2) to be more clear. - L190: I’m not sure if \lambda is used anywhere in the paper - L200: loose some property -> lose some properties - L201: depends -> depend - L219: the sentence “this theorem states …” is not clear, it should be rewritten - L319: Theorem -> theorem - L342: projections number -> number of projections Significance: I believe that the paper addresses an important question, which is of interest to the community. The resulting approach would be useful in many domains.

[Author Response · NeurIPS 2019]

Dear reviewers, we would first like to thank you for the helpful comments and suggestions of improvements. Remarks concerning typos, bad notations and missing references will be fixed according to your suggestions if the manuscript is accepted. Please see our detailed answer to your major concerns below:

**Computational aspects**: The implementation described in the paper corresponds to a naive one, that computes distances matrices along the projections, and is not the most efficient. Pykeops was used to avoid memory overflows in the evaluation of the cost in Eq (3) which was computed in $O(n^2)$ (both space and time). In fact this implementation was unnecessary since the final cost can actually be computed in $O(n \log(n))$. Indeed, one can develop the sum (3) to compute it in $O(n)$ operations: the term depending on $\sigma$ (Eq (2)) can be computed in $O(n)$ operations using $W(x, y, \sigma)$ as shown in the supplementary material and for the remaining constants $\sum_{i,j}(x_i - x_j)^4$ (idem for $y$) the binomial development gets rid of the $\sum_{i,j}$ and only involves $\sum_i$ terms that can be computed in $O(n)$ operations. Overall, in 1D, GW can be computed as efficiently as Wasserstein. As a consequence, the complexity of SGW is exactly the same as for Sliced

Figure 1: Runtimes comparison between SGW, GW, entropic-GW between 2D random distributions with varying number of points from 0 to $10^6$ in log-log scale

Wasserstein and Pykeops is not needed anymore. We believe this discussion can be added without changing the overall message of the paper by updating Fig 3 using a pure pytorch implementation as in Fig 1. With this implementation, one can compute SGW between 1M point distributions in 1s (vs 100s with a naive PyKeops implementation). Note also that entropic-GW is implemented on GPU as well. This way, it is clear that our method is responsible for the computational gain that **is not a consequence of using PyKeops** (#R1).

Related to the remark of (#R1) we also added the runtimes for two different numbers of projections $L = 50, 200$. The paragraph "Computational aspects" of the paper describes the influence of $L$ on the theoretical complexity. To the best of our knowledge, the effect of $L$ on the quality of estimation of the expectation is a hard question that is still open for the Sliced Wasserstein itself. Runtimes are computed between 2D measures since the dimension does not have an impact on the complexities for computing GW and SGW (they only depend on $L$ and $n$) (#R1). Moreover, the optimization over the Stiefel manifold does not depend on the number of points but only on the dimension $d$ of the problem so that overall complexity is $niter(Ln \log(n) + d^3)$, which is affordable for small $d$. On the spirals we observed that computing RISGW is one order of magnitude slower than the non-RI variant on CPU, which is still reasonable (# R1). We propose to add this discussion in the manuscript.

The non Hilbertian case is a very good remark (#R3). One straightforward approach would be to consider an embedding of the distances using multi-dimensional scaling as a prepossessing step or to learn distance-preserving embeddings using Siamese networks. This would come with an additional cost but we believe that this direction is worth investigating and will add it to the discussion.

**About the choice of** $\Delta$: The map $\Delta$ is one of the contributions of the paper. Here we propose a simple method (using a linear map in the Stiefel manifold) to align the spaces, even though one could consider other approaches (e.g a $\Delta$ parameterized by a neural network). We believe designing $\Delta$ is application dependant and preferred to restrict $\Delta$ to the Stiefel manifold in order to ensure rotation invariant guarantees so as to make the connection with an important property of GW. As such, we can use the $\Delta$ formulation when $p = q$ to recover this property (#R1). We thank (#R4) for its remark concerning the 4 others discrepancies. We originally did not want to add non obvious extensions of W, SW using the "$\Delta$ trick" and only focus our paper on Gromov-Wasserstein as our main result is Theo 3.1. We believe that using $\Delta$ with other discrepancies deserves a deeper study since it raises a lot of quantitative and theoretical questions (e.g closed forms for $\Delta$) and we chose not to include such discussions in the scope of the paper. From a purely computational point of view, complexity of all methods are the same (cf remark above). Nevertheless we will add these discrepancies and run corresponding experiments with them in the supplementary as suggested by the reviewer.

**Theoretical aspects and proofs**: (#R1) the problem (3) is equivalent to Eq (2) in 1D since the quadratic term is constant w.r.t. the permutation $\sigma$ (as being of the form $\sum_{i,j} c_{\sigma(i),\sigma(j)} = \sum_{i,j} c_{i,j}$) so that the minimization only involves the cross products. We will clarify this point.

**Experiments**: (# R1) The idea behind the simple spiral example was just to illustrate the different behaviors of GW, SGW and RISGW w.r.t rotations. Indeed, other rotation invariant methods could be applied here and would give similar results.

[Meta-Review · NeurIPS 2019]

This paper proposes a "sliced" approximation to the Gromov-Wasserstein distance (i.e., using random projections). The analysis builds on the closed-form solution of the GW problem in one dimension and then using the slicing trick to construct an estimator in the multidimensional case. A computational implementation in PyKeops compares favorably to existing methods. The consensus among the reviewers, taking into account the authors' response, is that this is a solid contribution to the literature on computational optimal transport.